

# Detection of atmospheric rivers affecting the Mediterranean and producing extreme rainfall over northern-central Italy

Silvio Davolio[1,2], Isacco Sala[1], Alessandro Comunian[1], Daniele Mastrangelo[2], Mario Marcello Miglietta[3], Lucia Drago Pitura[2,*], Federico Grazzini[4]

[1]Dipartimento di Scienze della Terra "A. Desio", Università degli Studi di Milano, Milan, 20133, Italy
[2]Institute of Atmospheric Sciences and Climate, National Research Council of Italy, CNR-ISAC, Bologna, 40129, Italy
[3]Institute of Atmospheric Sciences and Climate, National Research Council of Italy, CNR-ISAC, Padua, 35127, Italy
[4]ARPAE, Hydro-Meteorological and Climate Service, Bologna, 40122, Italy
[*]now at 3B Meteo S.r.l., Bergamo, 24036, Italy

*Correspondence to*: Silvio Davolio (silvio.davolio@unimi.it)

**Abstract.** Atmospheric rivers (ARs) have been recently identified also in the Mediterranean basin, and they have been shown to play an important role in intense precipitation events over northern Italy and the Alpine chain. In fact, as demonstrated by two recent severe events (27-29 October 2018; 02-03 October 2020), the synoptic pattern conducive to heavy rainfall in this area may favour an intense moisture transport from remote regions towards the Alps. In these events there was either a south-westerly moisture advection directly from the Atlantic Ocean Tropical area across the African continent, or a north-westerly transport over the Atlantic area, entering the Mediterranean in correspondence of the Gibraltar Strait.

In order to identify ARs in such a complex geographical area, a well-known algorithm of objective detection has been modified to take into account the peculiar morphology of the Mediterranean basin and, consequently, the peculiar shape of the organized water vapour transport, which may differ from that generally observed in the ARs over the open ocean. The two above-mentioned case studies have been used for testing the procedure.

Lastly, the algorithm has been applied in conjunction with some additional selection criteria for the identification only of the AR events that affected northern-central Italy in the last about 60-year. A climatological analysis is provided and the possible correspondence between the most intense identified ARs and extreme rainfall events is investigated. Exploiting a precipitation dataset for northern-central Italy (ArCIS), some areas turned out to be particularly exposed to extreme precipitation events in the presence of ARs.

## 1 Introduction

In the last decades, there has been growing interest to study the transport of relevant amounts of moisture from the (sub-) Tropics to the midlatitudes. This transport may sometimes be organized in the lower troposphere, assuming the form of narrow corridors named atmospheric rivers (ARs; Ralph et al., 2018). ARs play a key role in modulating the global water



cycle, being responsible for the majority of the water vapour transport at midlatitudes. Most importantly, ARs are strongly associated with heavy precipitation and floods in many regions around the world (Gimeno et al., 2014, for a short review). Since the pioneering study of Zhu and Newell (1998), ARs have been investigated especially over the eastern Pacific Ocean

and the U.S. West Coast (Neiman et al., 2011; Ralph et al., 2006; Rutz et al., 2014, among others). Here, several field campaigns over the past two decades took place (CALJET, Ralph et al., 2004; PACJET, Ralph et al., 2005; HMT, Ralph et al., 2013) and a reconnaissance program has been activated (Zheng et al., 2024), allowing to gain a deeper understanding of AR dynamics and characteristics, as well as to improve the forecasting of their impacts at different time scales.

In the last years, the impact of ARs has become evident in other parts of the globe (Gimeno et al., 2016) and the interest in
the topic has increased also over Europe. The link between ARs and extreme precipitation in Europe was first pointed out by Stohl et al. (2008) whose study, focussed on Norway, was followed by several papers dealing with other regions of western Europe: among others, Lavers et al. (2011) for the British Isles, Ramos et al. (2015) for the Iberian Peninsula, Benedict et al. (2019) for the Scandinavian Peninsula, Doiteau et al. (2021) for France. All these authors showed that AR landfall on the western European Atlantic coast produces extreme precipitation when an anomalous amount of moisture, coming mainly
from subtropical or tropical areas but also from mid-latitude regions, impinges in the form of an AR on the coastal orography and is forced to rise and condense. Lavers and Villarini (2013), Rössler et al. (2014), and Ionita et al. (2020) analysed the AR effects also over central Europe, revealing that the coastal orography of western Europe, relatively lower compared to that of the U.S. West coast, allows a deeper inland penetration.

Only a few studies have so far investigated the presence of ARs across the Mediterranean basin and their possible role in
modulating heavy rainfall over southern Europe and Italy in particular, although the importance of the large-scale moisture transport from regions outside the Mediterranean is known to be often associated with torrential precipitation (Insua-Costa et al., 2022). Some studies suggested a link between long-range transport of humidity and extreme precipitation (Malguzzi et al., 2006; Buzzi et al., 2014), but Krichak et al. (2015) were the first that applied a specific diagnostic, simply based on Integrated Vapour Transport (IVT) maps, to detect an AR during the infamous 1966 "century" flood in Florence (Italy).
More recently, Davolio et al. (2020) objectively detected the presence of an AR and investigated in detail its role, through an atmospheric water budget computation, during "Vaia" (Cavaleri et al., 2019; Giovannini et al., 2021), a major Mediterranean storm that caused extensive damages over the Alps. Lorente-Plazas et al. (2019) demonstrated the existence and importance of moisture transport through AR-like structures in the western Mediterranean and their hydrometeorological impact over the southern Spanish coast. Finally, ARs have been documented in some recent heavy precipitation events in the Mediterranean
(Martinković et al., 2017; Davolio et al., 2023) and in association with severe dust storms moving from the Sahara Desert towards Europe (Francis et al., 2022).

The Mediterranean basin is characterized by a complex morphology. Together with the presence of complex orography, often close to the coastal areas, this aspect may pose some complications in the application of detection methodologies and in the definition of AR landfalling. Therefore, the detection algorithms, which are generally suitable for open ocean areas, as
the Pacific or North Atlantic, may need some adjustments when applied to the Mediterranean. For example, vertically





integrated variables, such as the Integrated Water Vapour (IWV) or IVT, may be affected by the underlying orography causing discontinuities in objective detection algorithms, which are based on the exceedance of certain thresholds. Additionally, the identification of ARs from satellite may suffer from the variable surface background emissivity, which interferes with passive microwave retrievals (e.g., as for the Special Sensor Microwave/Imager (SSMI) sensors). Moreover, two recent heavy precipitation episodes over Italy, that will be used as a testbed in the present work, have shown that the origin of moisture can be either the tropical Africa or the North Atlantic. The different origin is associated with a very different geometry and shape of the ARs, sometimes jeopardizing automatic detection algorithms looking for objects typically oriented from southwest to northeast. Therefore, Mediterranean complexity requires a particular care, as also stressed by Lorente-Plazas et al. (2019).

The aim of the present study is to adapt a well-known AR detection algorithm (Guan and Waliser, 2015, GW15 hereinafter), largely used in the scientific literature, to the complexity of the Mediterranean basin (as described in Section 3), testing the results on two case studies previously analysed in recent research activities, and set up a methodology (Section 4) that allows to identify Mediterranean ARs affecting the target area of northern-central Italy. Finally, some statistics and results concerning the connection between ARs and extreme rainfall events are provided in Section 5.

## 2 Dataset and selected meteorological events

The ERA5 reanalysis dataset (Hersbach et al., 2020) is used in this study for both the synoptic analysis of the events and the AR detection procedure presented in the following section. For the latter, the longitudinal and meridional component of IVT, named IVT$_x$ and IVT$_y$ in the following, have been extracted at 0.25° resolution every 6 hours over the area 20°N – 60°N, 30°W – 30°E, for the period between 1961 and 2024.

This 64-year period, as well as the target area of northern-central Italy, have been selected considering the availability of a high-resolution (5 km × 5 km) gridded daily precipitation dataset named ArCIS. As described in Pavan et al. (2019), ArCIS is derived from the interpolation of a high-density surface observation network, consisting of more than 1700 raingauges distributed on 11 Italian regions, and includes also several stations of adjacent Alpine regions. Following a previous elaboration of the dataset (Grazzini et al., 2024), the daily precipitation is aggregated over area units of the Italian Department of Civil Protection, defined for the national operational warning system. These 94 areas aggregate homogeneous subregional hydrological basins of the territory (see Fig. 1 in Grazzini et al., 2020). The ArCIS dataset is exploited here to characterize the most intense events reported in the following sub-sections and in Section 5, and to investigate the possible correspondence between ARs and extreme precipitation events. The condition of extreme precipitation is attained when the aggregated observed daily precipitation of one or more of the 94 warning areas exceeds the 99th percentile of its wet-days climatology (with respect to the recent climate period, 1991–2020).





Two recent heavy precipitation episodes affecting northern and central Italy have been selected as testbeds since they were related to ARs with very different characteristics. They are briefly described in the following, but in-depth descriptions may be found in the references provided hereafter.

### 2.1 27-29 October 2018: storm "Vaia"

Between 27 and 29 October 2018, heavy rainfall, floods, storm surges and an extreme windstorm, associated with an explosive Mediterranean cyclogenesis, affected many parts of the Italian Peninsula, especially the north-eastern Alpine regions. Several recent papers (Cavaleri et al., 2019, Davolio et al., 2020, Giovannini et al., 2021) investigated the effects of this storm (also known as "Adrian"), comparable to the "century" flood of November 1966 (Malguzzi et al., 2006; Sioni et al., 2023) in terms of precipitation volume over northern-central Italy (Grazzini et al., 2020). A large amplitude baroclinic
wave developed over the North Atlantic and western Europe, favouring intense meridional heat and moisture exchanges. An upper-level trough deepened over the Iberian Peninsula and extended over North Africa as it slowly evolved eastward (Figs 5a-c in Davolio et al., 2020). Within this large-scale setting, typical of heavy rainfall over the Alps in autumn (Grazzini et al., 2021), cyclogenesis occurred over the western Mediterranean on 29 October, and a very intense surface low moved across the basin (Figs. 5d-f in Davolio et al., 2020), further intensifying the moisture transport coming from remote tropical
areas, as shown in Fig. 1a. The AR, together with an initial moisture input also from the Atlantic, contributed critically to heavy rainfall (Davolio et al., 2020; Sioni et al., 2023), reaching a local peak of almost 900 mm in 72 hours in the eastern Alps (Fig. 1c).

### 2.2 02-03 October 2020: storm "Alex"

Within a North Atlantic large-scale upper-level trough, a rapid cut-off process occurred at the beginning of October 2020
between UK and France (Fig. 2 in Davolio et al., 2023). The extra-tropical cyclone, named "Alex" by MeteoFrance on behalf of EUMETNET, but also known as "Brigitte", intensified very rapidly due to the upper level forcing of a very intense jet stream, and moved south-eastward, hitting the Britain coast on 02 October, where wind gusts exceeding 50 m s$^{-1}$ were recorded (Magnusson et al., 2021). During October 2, the cyclone, whose minimum mean sea level pressure reached 970 hPa, remained almost stationary over north-western France, while the associated cold front progressively swept over the
Mediterranean basin causing floods over Spain, France, and Italy. As shown in Fig. 1b, ahead of the cold front, an intense water vapor transport occurred, in the form of an impressive AR elongated from the North Atlantic, entering the Mediterranean basin through the Gibraltar Strait. The huge amount of moisture was responsible for heavy rainfall in Italy (Fig. 1d), over both the north-western coastal regions and the Alps (Davolio et al., 2023). Precipitation in excess of 300 mm affected a wide area of Liguria and Piedmont regions (for locations refer to Fig. 3), with local peaks above 500 mm in 12
hours. October 2 was classified as the rainiest day in Piedmont (in terms of average rainfall amount over the region) in the last 60 years with catastrophic consequences on the environment and infrastructures (Acordon and Cat Berro, 2024).



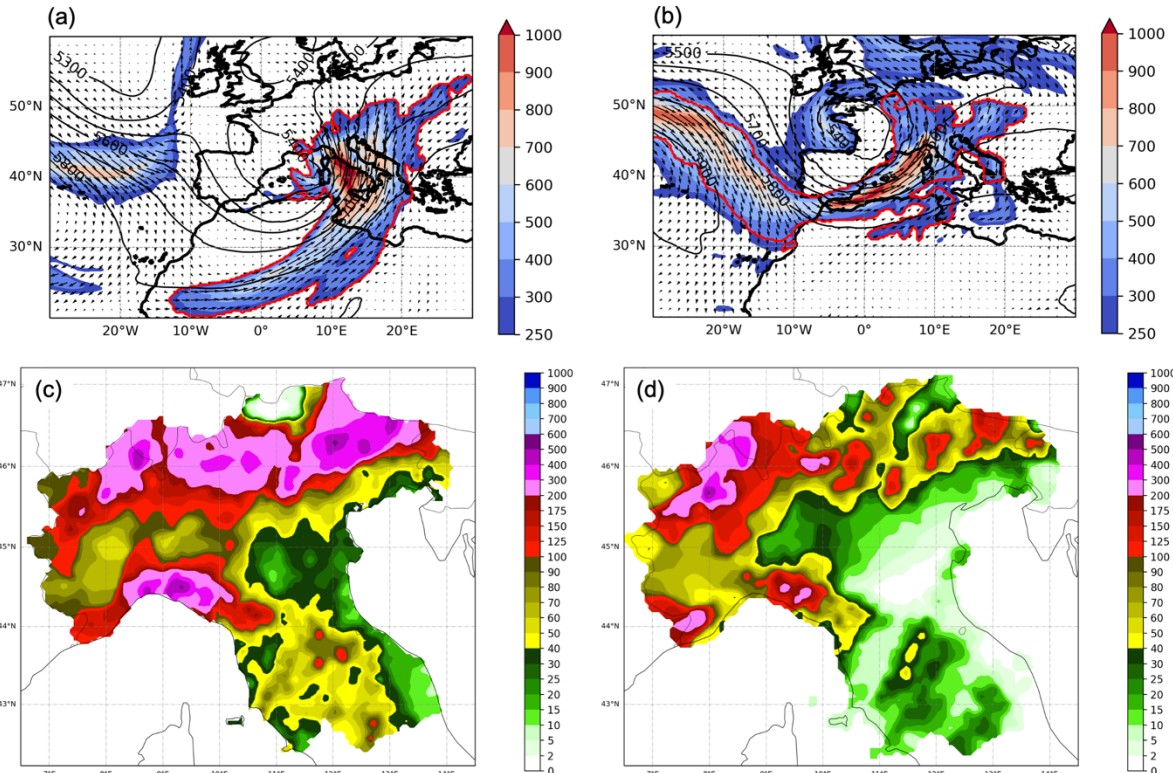

**Figure 1: 500 hPa geopotential height (m, black contour lines) and IVT (kg m⁻¹ s⁻¹, colour shading and vectors) for (a) the Vaia storm, at 12 UTC, 29 Oct. 2018 and for (b) Alex storm, at 18 UTC, 02 Oct. 2020. The red bold line surrounds the object identified as AR. Total precipitation (mm) accumulated in the two events: (c) in 72 h between 27 and 29 Oct. 2018 and (d) in 24 hours on 2 Oct. 2020. Precipitation data are extracted from the ArCIS dataset (Pavan et al., 2019).**

## 3 The detection algorithm

Since the number of studies dealing with ARs has expanded, many detection methods have been developed, differing in the selected variables, applied criteria, and input data (Shields et al., 2018). The first detection procedures were based on the total column IWV since this variable was easily available from polar satellite microwave retrievals. A feature characterized by IWV > 2 cm, length > 2000 km, and width < 1000 km was defined as AR by Ralph et al. (2004), who were the first to impose conditions for setting the boundaries of ARs, based on the CALJET field campaign results.

However, it became soon evident that the use of IVT, instead of IWV, permits a much more robust identification of a phenomenon that is characterized by both moisture and wind. Therefore, exploiting the availability of several global reanalysis datasets, most of the methodologies have been based on IVT, setting different thresholds (either fixed or climatologically based), together with geometric (shape) and temporal (persistence) criteria, possibly filtering out features that resemble ARs but are not. An IVT threshold of 250 kg m⁻¹ s⁻¹ has been typically adopted for midlatitudes (Rutz et al., 2014), although it may not be always appropriate (Reid et al., 2020), sometimes in combination with an IWV threshold (e.g., 15 mm in Gershunov et al., 2017). However, other studies applied variable IVT thresholds based on the percentiles of IVT





(e.g., 85th percentile, Lavers et al., 2012), a choice that makes it possible to diagnose an anomalous moisture transport in a specific area and season. The Atmospheric River Tracking Method Intercomparison Project (ARTMIP, Shields et al., 2018, Rutz et al., 2019) is an international effort aimed at quantifying uncertainties associated with different detection algorithms, possibly highlighting strengths and weaknesses. Interestingly, the choice of the reanalysis dataset turned out to impact the results less than the detection method adopted (Ralph et al., 2019).

Among the available algorithms, that of GW15 is one of the most widely used, and combines intensity and geometry thresholds for AR identification: i) a percentile-based threshold for IVT, depending on season and location, in combination with a fixed threshold ($100\ \mathrm{kg\ m^{-1}\ s^{-1}}$) that gets rid of climatologically weak-IVT areas; ii) shape requirements, i.e. AR length > 2000 km, and aspect ratio (length/width) > 2. Contiguous areas that satisfy these criteria identify objects candidate to be ARs, which then undergo other additional directional requirements, such as the direction of the object-mean IVT (poleward

component $\mathrm{IVT_y} > 50\ \mathrm{kg\ m^{-1}\ s^{-1}}$).

To apply the algorithm, IVT percentiles have been previously computed based on daily values at 12 UTC for the 30-year period ranging from 1991 to 2020. In particular, the 85th percentile has been defined for each month of the year as required in input by the GW15 algorithm.

Based on the problems detected in its application to the two case studies described in the previous section, changes have

been applied to the GW15 algorithm, of which we provide a step-by-step description in the following.

First, some preliminary sensitivity tests showed that the fixed IVT minimum threshold of $100\ \mathrm{kg\ m^{-1}\ s^{-1}}$ is not appropriate for the Mediterranean area. In fact, increasing this value to $250\ \mathrm{kg\ m^{-1}\ s^{-1}}$ allows to better identify AR-like structures in the area, emerging from the IVT background which is on average relatively high in the Mediterranean. However, the main failure of the GW15 algorithm concerns the detection of the AR during the October 2020 event, due to the complex U-

shaped structure (Fig. 1b) associated with the moisture transport: the moisture corridor develops from north-north-west to south-south-east all across the North Atlantic, moves south-eastward close to the western European coast, enters into the Mediterranean basin in correspondence of the Gibraltar Strait and finally moves north-eastward towards Italy. Since the GW15 algorithm was designed for poleward moisture transport ($\mathrm{IVT_y} > 0$), as it typically occurs over the Pacific Ocean, it misses to recognise this object as AR (Fig. 2a), although the IVT values largely exceed the prescribed thresholds.

However, if a southward transport of vapour is allowed, the algorithm identifies the object (Fig. 2b), but it also includes a narrow area of relatively high IVT wrapping around the extratropical cyclone, which is not part of the main humidity transport associated with the AR. To avoid the latter feature, one may simply impose that $\mathrm{IVT_x}$ must be positive (i.e. westerly). However, we found that this condition is not always appropriate. In fact, the 2018 event clearly shows (Fig. 2d) that, due to the cyclonic circulation over the Mediterranean, the AR transports moisture poleward but with a weak easterly

component. Therefore, in order to identify an object appropriate to represent the intense transport of humidity over the Mediterranean in both case studies, we finally decide to impose that the AR should be associated with a prevailing poleward transport. In other terms, the modified algorithm allows $\mathrm{IVT_x} < 0$ (westward), provided that its absolute value is less than that of $\mathrm{IVT_y}$ (Fig. 2c and 2d).





**Figure 2: AR detection attempts for the storms Alex (panels a, b, c) and Vaia (d). Colour shading indicates IVT (kg m⁻¹ s⁻¹) and starts from 250 kg m⁻¹ s⁻¹. The bold black line surrounds the object that is identified as an AR by progressive modification of the detection algorithm. In (a) the object is missing, since no AR is detected by the original algorithm; in (b) we have allowed southward transport, whilst (c) and (d) show the result of the modified algorithm applied in this study (see text for more details).**

While no problem comes up with the requirements of length (> 2000 km) and aspect ratio (> 2), for some of the 6-hour steps taken into consideration in the analysis of the ERA5 fields, the correct detection of the two ARs seems hampered by shape constraints. In the original algorithm, shape is linked to 'coherence' in IVT direction with respect to the object's mean IVT, and 'consistency' between object mean IVT direction and the overall orientation (GW15). After many trials and errors, undertaken by varying the pre-set values of these two parameters, the coherence is increased to 65° (instead of 45°), while the consistency check is discarded.

These modifications suitably adapt the algorithm to the very different characteristics of the two ARs within the Mediterranean area from those observed in the open oceans. Moreover, a recent release of the detection algorithm (Guan and Waliser, 2024, their Fig. 3 in particular) has included similar refinements, such as the possibility to detect zonal or even equatorward ARs.



Once verified that the approach is able to correctly detect the two selected case studies (as well as other selected events – not shown), it is applied to the full dataset, as described in the following Section.

## 4 Methodology for the identification of intense AR affecting northern-central Italy

The application of the algorithm to the 6-hourly ERA5 dataset over the target area (20°N – 60°N; 30°W – 30°E) for the period ranging from 1961 to 2024, identifies almost one thousand AR objects that satisfy the conditions defined in the previous Section. The aim of the present study is to identify only the ARs that affect the target area of northern-central Italy and group together those related to the same event. The selection procedure is described in the present Section.

Precipitation events over northern-central Italy are often characterized by warm and moist air advection over the Tyrrhenian and Adriatic Seas, taking the form of southerly low-level jet during severe events (Buzzi and Foschini, 1998; Miglietta and Davolio, 2022) conveying water vapour towards the Apennines and the Alpine slopes. It is not rare that elongated corridors of moisture transport appear in the Mediterranean, although too short to be considered ARs. Sometimes, the identification of the axis of these objects produces an overestimation of their length, since the algorithm may detect a zigzag axis instead of an almost straight line. Thus, it may happen that "false" ARs are revealed. In order to distinguish between this intense but local transport from the Mediterranean basin and the vapour really coming from remote sources, and to select only ARs landing on the target area, one additional condition is required: the shape of the AR object identified by the detection algorithm must cover at the same time both the target area over northern-central Italy and a remote source region outside the Mediterranean (Fig. 3). The latter is devised to intercept ARs reaching the Mediterranean from either Africa tropical areas or the North Atlantic, as in the two selected case studies, and assures a length of at least 2000 km.

Also, a temporal requirement is imposed in the AR detection, that is the AR must cover the target area and the remote source regions in Fig. 3a for at least three 6-hourly time steps. It means that only AR objects persisting longer than 12 hours are considered as AR events, as set also in previous studies (Ramos et al., 2015; Gershunov et al., 2017; Lorente Plazas et al., 2019).

The whole procedure provides the AR shape and axis in a netcdf file. Moreover, in a text file, the time/data of start and end of each event is provided. Unfortunately, since the Mediterranean is an enclosed basin, entirely surrounded by land areas and characterized by several islands, the algorithm cannot provide the exact position of the landfall over the Italian peninsula, which would assume only one sea-land transition along the AR path. In order to provide a measure of the intensity of the AR, the maximum value of IVT during the event, attained within the object shape over the sea in proximity of the Italian coast, is selected and reported in the same file.

Finally, a careful check of all the selected events was performed to manually exclude few events in which the detection was due to a partial failure of the algorithm in computing the geometrical requirements or to AR objects clearly flowing to the north of the Alps, but with some marginal or residual object portion slightly overlapping the target area. Following this procedure, a dataset of ARs affecting northern Italy is now available.



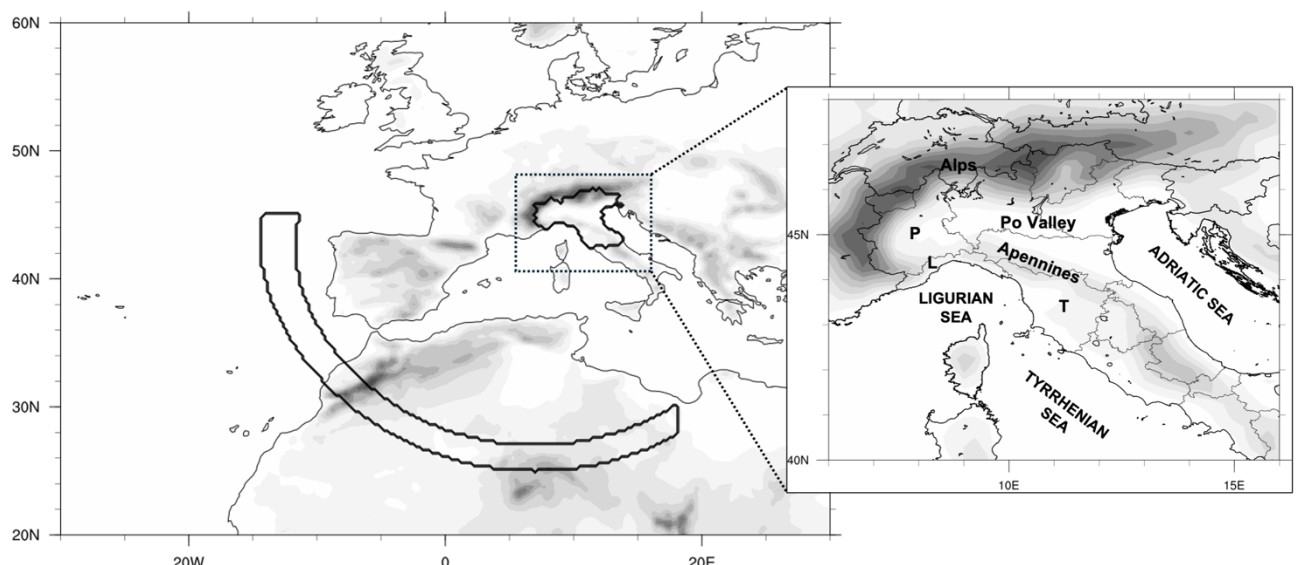

**Figure 3: Domain considered for the identification of Mediterranean ARs. Shading represents orography elevation every 250 m. The administrative area of northern-central Italy is surrounded by a bold line, and it represents the target area where precipitation observations (ArCIS) are available. The arc-shaped area, together with the target area, must be covered at the same time by the shape of the AR, in order to select the AR for our analysis. Locations cited in the text are indicated on the zoomed map. L, P and T indicate Liguria, Piedmont and Tuscany region, respectively.**

## 5 Results

A total of 357 AR events affecting northern-central Italy are identified between 1961 and 2024, with an average of 5.6 AR per year (Fig. 4). Two events are considered independent if separated at least by one day. During 1987 a maximum of 13 events was attained, whilst in 1989, 2005 and 2007 only one event occurred.

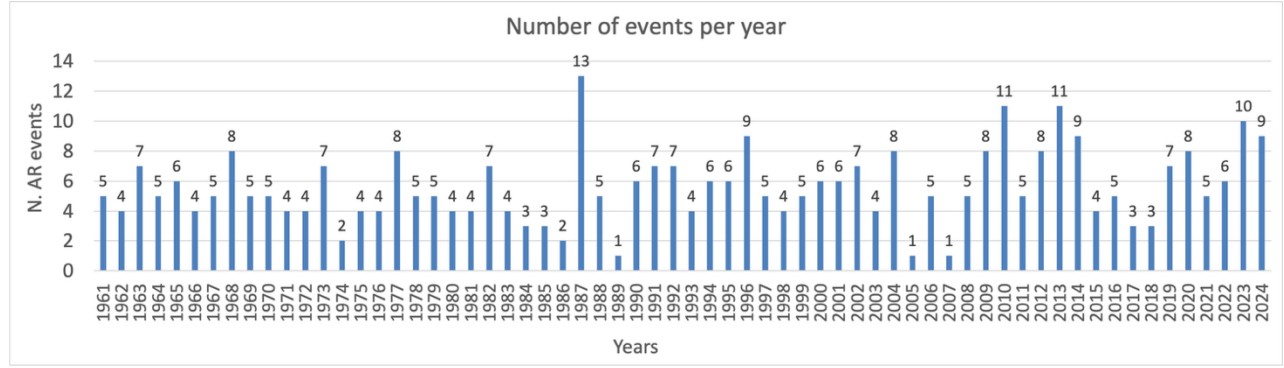

**Figure 4: Number of AR events affecting northern-central Italy each year, in the period 1961- 2024**

In terms of intensity and duration, Fig. 5a shows that about half of the events end within 24 hours, and almost 90% within 48 hours. It is worth recalling that, following the criteria defined in Section 4, the lifetime is defined as the period during which the shape of the AR, as provided by the detection algorithm, overlaps both the target area of northern-central Italy and the





curved area outside the Mediterranean. Therefore, it might be possible that the AR persists longer over the Mediterranean basin besides this period. The maximum IVT distribution is centred around 600 kg m$^{-1}$ s$^{-1}$, with some remarkable events that exceed 1000 kg m$^{-1}$ s$^{-1}$ (Fig. 5b).

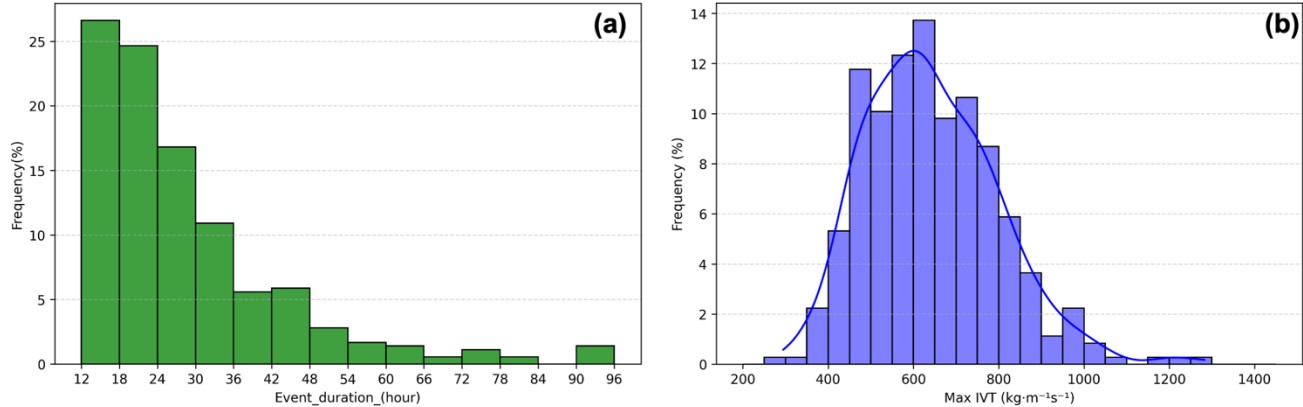

**Figure 5: Frequency distribution of (a) duration of AR events and (b) AR intensity, defined as max IVT attained close to the Italian peninsula coastline.**

The monthly distribution of AR events as a function of their intensity is provided in Fig. 6a, where AR classification is only based on maximum IVT: weak (between 250 kg m$^{-1}$ s$^{-1}$ and 500 kg m$^{-1}$ s$^{-1}$), moderate (500 - 750 kg m$^{-1}$ s$^{-1}$), strong (750 - 1000 kg m$^{-1}$ s$^{-1}$), extreme (1000 - 1250 kg m$^{-1}$ s$^{-1}$), and exceptional for larger values. Adopting a more sophisticated classification proposed by Ralph et al. (2018), that merges the maximum IVT and the duration, five AR levels are defined (from AR1 to AR5) and their monthly distribution is presented in Fig. 6b.

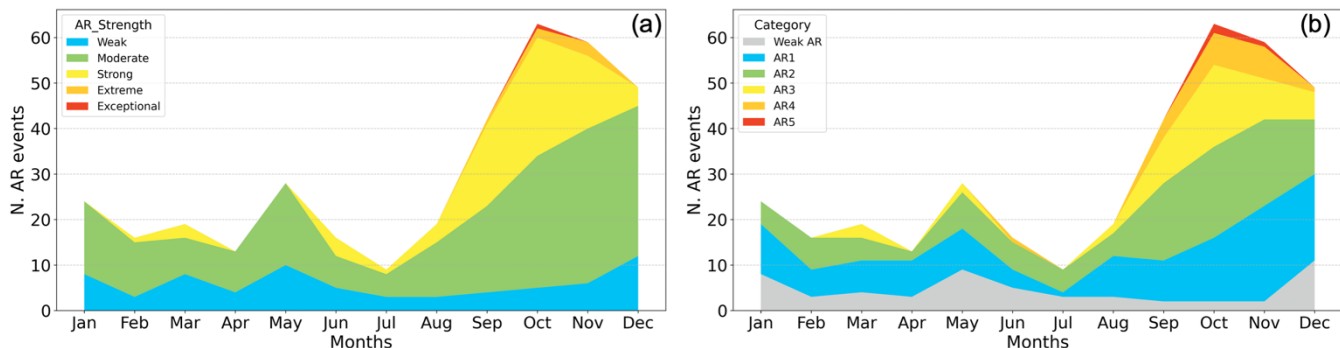

**Figure 6: Monthly distribution of the number of ARs for the entire period 1961-2024, following a classification based on max IVT only (a) or merging max IVT and duration (b) as proposed by Ralph et al. (2018).**

Regardless the adopted classification, Fig. 6 shows clearly that autumn-early winter is the period of the year most favourable to AR in the area, while a secondary much weaker peak appears in spring. Also, autumn is the only season that experiences the presence of very intense ARs belonging to the two highest categories. This result agrees with the findings of Grazzini et al. (2020) that studied extreme precipitation over the same target area. They found that, during extreme precipitation events, the intensity of the meridional component of IVT presents a behaviour similar to the AR frequency shown in Fig. 6, with a



clear maximum in autumn due to both enhanced moisture availability and a favourable phasing of the upper-level circulation. This is an indication that ARs can be strictly connected to extreme rainfall in northern-central Italy.

A preliminary analysis ranked the identified AR events based on the associated maximum IVT values (Table 1). Interestingly, both the selected events discussed in Section 2 are included in this ranking (2018 Vaia storm as second and 2020 Alex storm as sixth), as well as the well-known century flood of 1966 (Malguzzi et al., 2006; Sioni et al., 2023) that

severely affected Tuscany region (particularly the city of Florence) and the north-eastern Italian Alps, including the city of Venice (De Zolt et al., 2006).

The most intense AR event in the list affected mainly southeastern France and the western Alps and only briefly the Italian peninsula during its strongest phase. Being strongly west-east oriented, most of the precipitation occurred over the Alpine divide, but especially on the French side (MeteoFrance, 2023). In any case, extreme precipitation and floods were reported

also over Italy (ARPA Piemonte, 2023).

| INIT DATE | DURATION | MAX IVT | CLASSIFICATION | CATEGORY |
|-----------|----------|---------|----------------|----------|
| 2023/10/19 - 00 UTC | 54 | 1282 | Exceptional | AR5 |
| 2018/10/27 - 06 UTC | 66 | 1208 | Extreme | AR5 |
| 1966/11/03 - 12 UTC | 24 | 1195 | Extreme | AR4 |
| 2012/11/03 - 12 UTC | 48 | 1056 | Extreme | AR5 |
| 1999/09/19 - 12 UTC | 18 | 1047 | Extreme | AR3 |
| 2020/10/02 - 00 UTC | 24 | 1028 | Extreme | AR4 |
| 2023/11/02 - 06 UTC | 24 | 1010 | Extreme | AR4 |

**Table 1: The 7 most intense AR over the Mediterranean, affecting northern Italy between 1961-2024.**

The 2012 event was monitored during the Special Observing Period of the HYMEX field campaign (Ducrocq et al., 2014). The AR, driven by a smooth trough over the western Mediterranean, induced heavy rainfall mainly over Liguria and Tuscany (Intensive Observation Period 19, see Ferretti et al., 2014 for further details), with convection over the Ligurian Sea

and orographic precipitation along the Maritime Alps.

For the 1999 event, short and intense precipitation was reported (Provincia Autonoma TN, 1999) together with minor floods and bank erosion in the central Alpine area.

Finally, the November 2023 event was characterized by major floods in Tuscany (LaMMA, 2023) with extreme precipitation (return time estimated around 50 years) that affected also the Apennine divide producing local floods, soil erosion and

landslides in the northern side of the mountain chain (ARPAE, 2023).



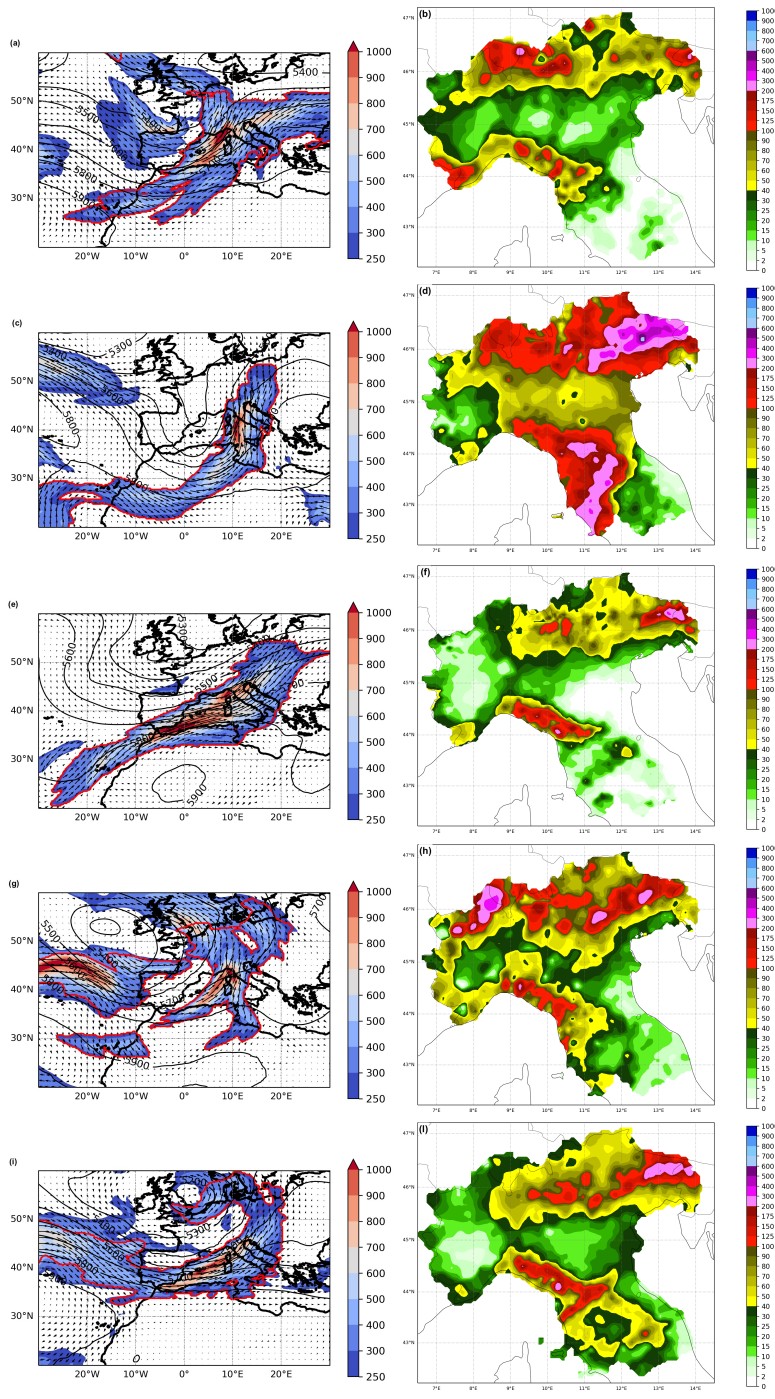

**Figure 7: The most intense AR events in terms of max IVT in the catalogue, as reported in Table 1. Left column: IVT (kg m-1 s-1, shading and vectors), geopotential height at 500 hPa (m) and the AR object identified by the bold red line; right column: accumulated precipitation (mm in 48h) for each event. Note that two intense events were already shown in Fig. 1. (a) 20 Oct. 2023, 06 UTC, (c) 04 Nov. 1966, 00 UTC, (e) 05 Nov. 2012, 00 UTC, (g) 20 Sep. 1999, 00 UTC and (i) 02 Nov. 2023, 18 UTC.**




The most intense ARs are associated with very intense precipitation events and severe impacts on the affected areas, thus indicating that extreme transport of humidity from remote areas leads to extreme rainfall and can represent an important diagnostic to take into account for forecasting purposes.

To further investigate the nexus between extreme precipitation and ARs, following the methodology proposed by Lavers and
Villarini (2013) for Europe, the top ten extreme precipitation events have been extracted for each warning area. Figure 8 shows the number of these events associated with an AR. Over most of the Alpine area, more than half of extreme events are caused by an AR. Some regions in the north-western and in the north-eastern Alpine sectors present 8 out of 10 events connected with AR. Similarly, the mountain chain of Liguria region, close to the coastal line, and the region close to the northern Apennine crest reaching elevation up to about 2000 m, are strongly affected by the impact of ARs. Conversely, the
Po Valley, which lays in the lee of the Apennines with respect to the prevailing westerly or south-westerly currents, the northern slopes of the Apennines themselves, as well as the Adriatic areas, appear to be shielded by the orographic effect and ARs are clearly not the main cause of extreme events. Moving south over central Italy, where the orography does not attain very high elevations and AR are not frequently perpendicular to the mountain axis, ARs present some impact, although they do not dominate the upper tail of the rainfall distribution.

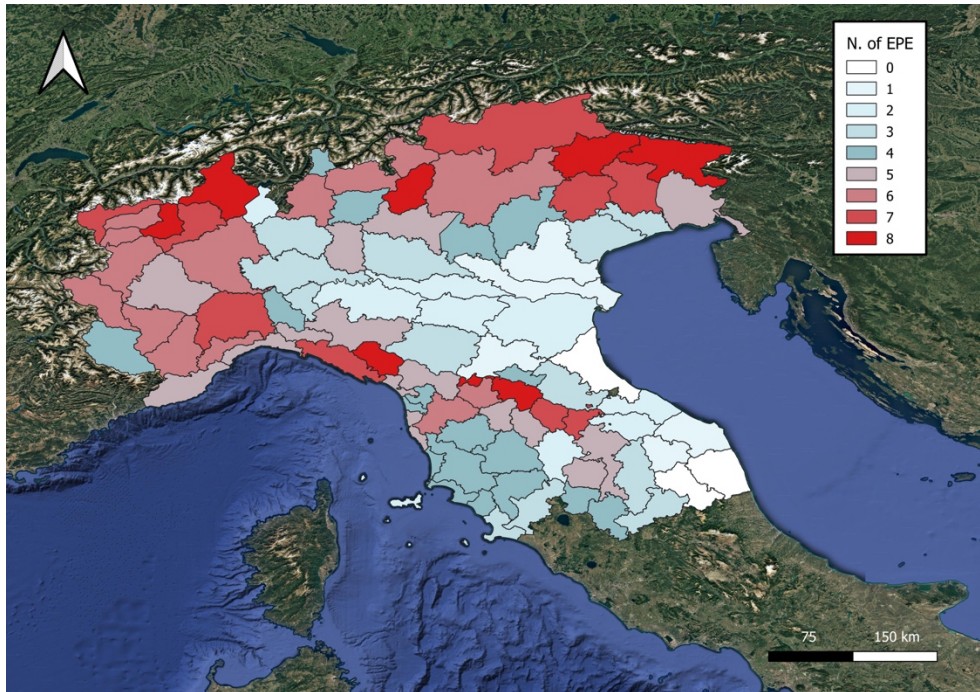


**Figure 8: Number of the ten most intense extreme precipitation events, for each warning area, that are associated with the presence of an AR. Red shading indicates a percentage greater or equal than 50% (figure made with QGIS).**



## 6 Conclusions

The main aim of the present study was to define a procedure to identify ARs entering the Mediterranean and to select those affecting the target area of northern-central Italy. The procedure is based on a well-known detection algorithm, suitably modified to manage the complexity of the Mediterranean region and the peculiarity of the AR events. After a careful testing phase, the detection methodology has been applied to ERA5 reanalysis for the period 1961-2024 and a dataset of AR events, including dates, duration, maximum IVT, shape and graphical products, has been produced and allowed a climatological analysis.

The main characteristics of the 357 AR events over the Mediterranean and affecting the target area of northern-central Italy have been identified. In particular, a well-defined seasonality emerged, with not only the majority but also the most intense ARs occurring in autumn-early winter period. This result is in line with Grazzini et al. (2020) that found the anomalous meridional IVT as a common ingredient for extreme precipitation over northern Italy, without discriminating between local transport or remote source contribution. Our results reveal that ARs contribute markedly to this picture by transporting moisture from outside the Mediterranean and also indicate a clear and strong connection between intense horizontal water vapour transport and heavy precipitation over the orography. In fact, extreme precipitation is dominated by the presence of ARs on several regions in the Alpine or Apennines areas.

The enhanced predictability of the large-scale water vapour transport has the potential to extend the lead time of extreme rainfall forecasts: recently, it has been shown (De Florio et al., 2019; Huang et al., 2021; Reid et al., 2024) that, up to the subseasonal time scale, IVT can be an effective predictor for extreme precipitation forecasting in regions where intense moisture transport is strictly connected with rainfall generation. Therefore, having shown that the presence of ARs is a critical factor for several civil protection warning areas in our target domain, the findings of this study result potentially important for early warning applications. Although limited to a few case studies of major floods over the Alps, interesting results are emerging along this line (Mastrangelo et al., 2025) and the study of ARs in the Mediterranean seems promising.

Of course, this study establishes only the basis for further and more comprehensive analyses on the Mediterranean area that are still missing in the literature, i.e. a systematic evaluation of the contribution of ARs to the overall precipitation, a more detailed analysis of the seasonal behaviour and the identification of possible connections between AR and Mediterranean weather regimes.

**Code and data availability**

ERA5 reanalysis data were downloaded from the Copernicus Climate Change Service https://doi.org/10.24381/cds.143582cf (Hersbach et al., 2020). The GW2015 algorithm was provided upon request by Bin Guan (bin.guan@jpl.nasa.gov). The final dataset in netcdf format is available upon request to the corresponding author.

**Author contributions**



Conceptualisation, SD, MMM; methodology and software, SD, IS, AC and LDP; analysis, SD, IS, LDP, MMM, DM; writing – original draft preparation, SD, MMM, DM; writing – review and editing, SD, IS, AC, DM, MMM, LDP, FG;

supervision and funding acquisition, SD and DM All authors have read and agreed to the published version of the paper.

**Financial support**

Financial support is acknowledged from Next Generation EU, Mission 4, Component 1, CUP B53D23007490006, project "Exploring Atmospheric Rivers in the Mediterranean and their connection with extreme hydrometeorological events over

Italy: observation, modelling and impacts (ARMEX)". This study has been also supported by the project "FOE2019: Cambiamento climatico: mitigazione del rischio per uno sviluppo sostenibile", funded by the Italian Ministry of University and Research, and the by the Civil Protection of Italy under the contract "DPC 2020-2021- Accordo DPC/CNR–ISAC".

**Acknowledgments**

The authors gratefully acknowledge the Italian Civil Protection Department for providing the shape files of the warning area units and the management of the ArCIS dataset for allowing the access to the data.

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
