# Peer review of "Detection of atmospheric rivers affecting the Mediterranean and producing extreme rainfall over northern-central Italy"

_EGUsphere, 2025_

## Referee Comment (RC1)

**Comments to authors (egusphere-2025-3447)**

In this manuscript, the authors explore the climatology of Atmospheric Rivers (ARs) entering the western Mediterranean and making landfall in northern–central Italy. Meteorological events with this configuration are illustrated through two case studies, "Vaia" and "Alex," which occurred in October 2018 and 2020, respectively, and led to extreme precipitation in northern Italy. The authors tune a state-of-the-art AR detection and tracking algorithm to develop a climatology (1961–2024) of events with these characteristics, identifying a clear seasonality: ARs entering the western Mediterranean and making landfall are most frequent and most extreme in autumn and early winter. Additionally, the authors use the ArCIS precipitation dataset to assess the contribution of North Atlantic AR intrusions to precipitation extremes in northern–central Italy, finding that up to eight of the ten most extreme precipitation events in the Italian Alps and Apennines are associated with ARs.

Overall, the manuscript is well written, and the results are relevant and interesting, with the potential to attract considerable attention from readers of this journal. I recommend a few improvements and clarifications prior to publication.

**Comments:**

1. This manuscript focuses on a specific type of AR intrusion from the North Atlantic into the western Mediterranean, reaching northern–central Italy. While this focus is generally well explained throughout the manuscript, it may be helpful to occasionally remind the reader that these events are not representative of ARs across the entire Mediterranean basin (e.g., line 327). I would suggest considering the addition of "the western Mediterranean" to the title. Moreover, it might be beneficial to ensure that this scope is stated consistently throughout the manuscript, including in the abstract, for example as it is clearly expressed in the first sentence of the Conclusions (lines 306–307).

2. Other studies reported similar behavior of ARs reaching eastern Mediterranean and middle east, I think those could be included in the introduction (lines 49-61) as these investigate the eastern side of the same Mediterranean basin. For example:
   - *Francis, D., Fonseca, R., Bozkurt, D., Nelli, N., & Guan, B. (2024). Atmospheric river rapids and their role in the extreme rainfall event of April 2023 in the Middle East. Geophysical Research Letters, 51, e2024GL109446. https://doi.org/10.1029/2024GL109446*
   - *Ezber, Y., Bozkurt, D. & Sen, O.L. Impact of atmospheric rivers on the winter snowpack in the headwaters of Euphrates-Tigris basin. Clim Dyn 62, 7095–7110 (2024). https://doi.org/10.1007/s00382-024-07267-2*

3. Could the authors clarify why extreme precipitation events are selected using the 99th percentile computed only over the climatic period 1991–2020, rather than over the full dataset period (1961–2024)? If extreme precipitation has increased over recent decades, using the 99th percentile based on the last 30 years could potentially lead to missing extreme events from the earlier part of the record. An alternative approach could be to de-trend the time series and then identify extreme precipitation events

based on the de-trended data. Even if the differences in the selected extreme events are small, please ensure that this choice does not affect the robustness of the results.

4.  In lines 156-157 similar concern arises on why the IVT 85[th] percentile is calculated using the period of 1991 to 2020 and not the full length of the data. As before please make sure this not affects your results.

5.  In lines 187–189, does this mean that the difference between the two directions (mean IVT and object orientation) is allowed to be as large as 65°? The wording was somewhat confusing when referring to "coherence." Given that these adjustments to the GW15 algorithm are extremely valuable for the scientific community interested in applying AR detection methods to regions with complex orography, it would be helpful to clarify them as much as possible. The rest of the modifications applied are reasonable, have been tested and seem to work well.

6.  In lines 219-221, it is not clear where the maximum-IVT is measured and how this location is selected. Is it a fixed point in the Ligurian sea for all ARs?

7.  Together with Figure 3, I would have appreciated seeing the AR frequency climatology for ARs entering the western Mediterranean and reaching northern–central Italy. This would allow for a quick assessment of the AR detection methodology proposed in the manuscript and facilitate comparison with other algorithms. For instance, based on my experience, detecting ARs using global detection algorithms within a limited domain (20–60° N; 30° W–30° E) can sometimes lead to missed ARs near the domain boundaries, as those might be partially outside of the domain. This may not be a major issue in the present case, since only ARs that reach northern Italy and the arc-shaped region are considered; nevertheless, including the AR frequency would still be useful to visualize the spatial distribution of the detected events.

8.  Figure 4 shows the number of events per year, and from a visual inspection there appears to be a possible positive trend. Could the authors clarify whether this trend reflects a warming climate effect or whether it might instead be an artifact of defining the IVT threshold using the 85th percentile from the most recent period? It would be helpful to verify whether this behavior is sensitive to the AR selection methodology (see my comment #4), and, if not, to assess and report whether the trend is statistically significant.

9.  In line 250, the AR scale by Ralph et al. (2018) is used. It would be helpful to acknowledge that this scale was originally developed for the west coast of the United States and is therefore not necessarily tailored to ARs making landfall within the Mediterranean basin. Nevertheless, the scale remains useful and relevant for presenting the results shown here. One possible option could be to apply the scale to the arc-shaped area outside the Mediterranean in order to assess the intensity of the ARs before they enter the basin.

10. In line 308, it might be worthwhile to briefly restate the main modifications made to the GW15 algorithm for application in the Mediterranean basin. I believe this represents an important outcome of the study and would be well suited for inclusion in the Conclusions.

11. In line 326, the manuscript refers to the interesting results from Mastrangelo et al. (2025). It might be helpful to briefly summarize the nature of these results for the reader. In addition, since the current reference appears to point to a conference abstract, the authors could consider citing a preprint of this work, if available.

---

## Referee Comment (RC2)

**Review of "Detection of atmospheric rivers affecting the Mediterranean and producing extreme rainfall over northern-central Italy"**

**Recommendation: Accept with major changes.**

This study aims to address two research gaps:

**RG1:** Mediterranean atmospheric rivers (ARs) exhibit geometrical and dynamical properties that differ from those of ARs over open oceans, potentially challenging automatic detection algorithms.

**RG2:** Only a few studies have so far investigated the presence of ARs across the Mediterranean basin and their potential role in modulating heavy rainfall over southern Europe, and Italy in particular.

To address these gaps, the authors build on tARget v1 (Guan & Waliser, 2015), a previously developed AR identification algorithm. The algorithm is refined to effectively detect ARs transporting moisture from the Atlantic into the Mediterranean and making landfall over northern–central Italy. The detection methodology is illustrated through two case studies, storms *Vaia* and *Alex* (October 2018 and October 2020, respectively), both associated with extreme precipitation over Italy. The authors then compile an AR database to characterize the seasonality of these events and to quantify their contribution to extreme precipitation in northern–central Italy over the period 1961–2024.

I find this study interesting and timely. The manuscript is clearly written and well structured, the figures are informative, and the conclusions are consistent with the evidence presented and directly address the stated research questions. I particularly appreciate the authors' use of state-of-the-art datasets tailored for the study region, such as the ArCIS precipitation product and the area units of the Italian Department of Civil Protection defined for the national operational warning system. These choices facilitate effective knowledge transfer to policymakers and other stakeholders.

I believe this paper will be of interest to the readership of *NHESS*. The analysis presented is rigorous and I agree with the findings. I nevertheless have some concerns regarding the novelty of the proposed detection algorithm and the application of the AR strength scale introduced by Ralph et al. (2019). AR science is a rapidly evolving field, and recent methodological developments appear to have addressed, at least in part, the first research gap identified by the authors. In addition, certain nuances related to the application of the AR scale from a Lagrangian instead of a Eulerian perspective may warrant further consideration. I elaborate on these points in the major comments below and offer specific suggestions for the authors' consideration.

**Major comments**

**Comment 1: Choice of AR detection algorithm and novelty**

In the introduction, the authors acknowledge the wide range of AR detection algorithms currently available at both regional and global scales, based on a variety of methodological approaches. They state that existing state-of-the-art algorithms, including tARget by Guan & Waliser, may not be well suited to identifying Mediterranean ARs, which motivates their decision to develop a regional AR catalog based on modifications to tARget.

The manuscript builds on tARget v1, originally developed in 2015. However, tARget has since undergone several substantial developments and is currently available in version 4, released in May 2024 (Guan & Waliser, 2024). My main concern is that several of the modifications introduced by the

authors to tARget v1 appear to already be implemented in tARget v4. It is therefore unclear why the first version of the algorithm was chosen instead of the most updated one.

This issue is illustrated by the case studies of storms *Vaia* and *Alex*. The authors motivate their methodological choices by stating that tARget fails to detect the AR associated with storm *Alex* (Figure 2 of the manuscript). However, tARget v4 detects the ARs associated with both *Alex* and *Vaia*. To support this statement, I provide Figure 1 of this review, which shows the ARs detected by tARget v4 during these two events (panels a and c).

Given the rapid evolution of AR detection methodologies, I encourage the authors to more clearly position their approach relative to recent developments in the field. In particular, it would be important to clarify which methodological gaps remain unaddressed by current state-of-the-art algorithms and how the proposed modifications advance beyond existing implementations.

[Figure]

**Figure 1.** Detection of AR contour and axis for (a, b) storm Vaia, at 12 UTC, 29 Oct. 2018 and for (c, d) storm Alex, at 18 UTC, 02 Oct. 2020, employing (a, c) tARget v4 and (b, d) PIKART v1 algorithms. The color shading indicates the IVT magnitude. The cyan triangle indicates the land-falling location identified by PIKART v1

**Comment 2: AR tracking methodology and event selection**

Beyond AR detection, it appears that the authors have implemented an AR tracking algorithm to connect individual AR contours into trajectories. However, the description of this tracking methodology is not sufficiently detailed to allow proper evaluation or reproducibility. From the manuscript, I infer that an overlap-based approach was used, but it is unclear how mergers and separations are handled. These choices directly affect key AR properties such as lifetime, spatial extent, and persistence.

Related to this point, the authors based their analysis on ARs that simultaneously intersect the Atlantic Ocean and Italy, based on the overlap between the AR footprint and a circular sector (Figure 3 of the manuscript). It is unclear how sensitive the results are to the location and geometry of this circular sector. Moreover, this criterion may exclude ARs that originate over the Atlantic and reach the study region without fulfilling this condition at a given time.

Several modern AR detection and tracking algorithms, including tARget v4, explicitly track ARs from origin to termination. Exploiting these advancements could reduce the need for additional parametrization and provide a more physically grounded selection of ARs based on the evolution of their trajectories rather than on instantaneous spatial intersections.

**Comment 3: Uncertainty associated with AR detection and tracking**

The authors thoughtfully acknowledge that AR detection and tracking represent one of the main sources of uncertainty in AR science, as underscored by the ARTMIP project and related studies. Despite this, the results presented in the manuscript rely on a single algorithm. Consequently, the manuscript does not provide an assessment of the uncertainty associated with methodological choices in AR detection and tracking. Given that the key conclusions rely on the identified AR climatology and its linkage to extreme precipitation, an evaluation of methodological uncertainty is essential to support the robustness of the findings.

Reproducing the main results with at least one additional AR detection algorithm would allow the authors to assess the sensitivity of their conclusions to the detection methodology. If the AR signal over northern–central Italy is physically robust, its main climatological features and impacts should be relatively insensitive to the specific detection algorithm employed.

**Comment 4: AR landfall and recent methodological advances**

In the introduction (lines 62–64), the authors emphasize the complexity of Mediterranean morphology and orography as a key challenge for AR detection and for defining AR landfall. Later (lines 217–219), they note that their algorithm cannot provide an exact landfall location over the Italian peninsula because it assumes a single sea–land transition along the AR path.

This is indeed a fundamental challenge in the Mediterranean context. It has also motivated the development of alternative AR detection approaches. For instance, the PIKART algorithm (published in August 2025) adopts a fundamentally different detection strategy than tARget and is able to identify AR landfall locations even in complex regions such as the Mediterranean. I verified that PIKART successfully detects the ARs associated with storms *Vaia* and *Alex*, including their landfall locations, as shown in Figure 1b,d of this review. The PIKART catalog and code are publicly available.

I fully acknowledge that the results presented in this manuscript were produced prior to the publication of PIKART, and I do not demand reprocessing the analysis using this algorithm. However, I recommend updating the introduction to reflect recent developments in AR detection for complex

regions and considering whether aspects of the PIKART approach could inform future refinements of the proposed regional catalog, particularly regarding landfall identification.

**Comment 5: Use of the AR strength scale**

The manuscript applies the AR strength scale proposed by Ralph et al. (2019) to characterize the intensity and persistence of ARs. Based on the description in lines 219–221 and 239–241, it appears that the scale is applied to individual AR objects identified by the detection algorithm, assigning a rank to each AR.

This approach raises concerns regarding the interpretation of the AR strength scale. The scale is designed to rank AR conditions at fixed locations using a Eulerian framework, rather than to classify ARs themselves from a Lagrangian perspective. In other words, it evaluates atmospheric conditions experienced at a given location during AR passage, not the intrinsic strength of a detected AR object.

This nuance is critical and has been discussed in previous studies (e.g. Guan et al., 2023; see also https://cw3e.ucsd.edu/arscale/). To align with the intended use of the AR strength scale, the authors should revise their methodology, for example by evaluating AR conditions at each grid cell within the study region or at selected reference locations, as currently implemented in operational forecasting systems such as the one by the CW3E along transects of the west coast of the United States.

**Recommendations**

As illustrated in Figure 1 of this review, tARget v4 and PIKART v1 provide alternative approaches to AR detection based on more permissive or restrictive criteria. To strengthen the manuscript while preserving its originality, I provide the following suggestions:

1. **Focus on AR climatology and impacts over northern-central Italy:** Rather than developing a new AR detection algorithm, I suggest that the authors consider using two or more of the freely available, state-of-the-art AR catalogs as the basis for their study. This would allow the manuscript to focus on the climatology, impacts, and associated uncertainties of ARs affecting northern–central Italy, while avoiding the additional complexity of developing a new detection method.
2. **Use of the most updated version of tARget:** If the authors consider that a new regional catalog is still needed for this study, I recommend using tARget v4 as the baseline, rather than version 1. Modifications should focus on aspects not already implemented in version 4, for example, adjusting the lower limit of IVT magnitude to 250 kg m⁻¹ s⁻¹. This would require updating the corresponding results and, in particular, Figure 2 of the manuscript.
3. **Documentation of differences:** The manuscript should provide a thorough comparison between the original tARget algorithm and the new regional catalog. Beyond showing AR contours for two case studies, the authors should quantify differences in key aspects that affect uncertainty, such as the number of events, frequency, intensity, and persistence of AR conditions over the study region (see, for example, Figure 7 of Vallejo-Bernal & Braun et al., 2025).
4. **Assessment of methodological uncertainty:** To evaluate the robustness of the results, I suggest applying at least one additional AR detection algorithm to reproduce the main findings. If the AR signal in northern–central Italy is physically robust, the main climatology and associated impacts should remain consistent across detection methods.
5. **AR strength scale methodology:** The authors should clarify how they apply the AR strength scale of Ralph et al. (2019). If the current approach assigns ranks to individual AR objects, it should be revised, as the scale is intended to rank AR conditions at specific locations (Eulerian approach), rather than ARs themselves (Lagrangian approach).

**Minor comments**

**Comment 1:** I have the impression that the colormaps used in Figures 1, 2, and 7 may not be fully colorblind friendly. I encourage the authors to verify this point and, if needed, make appropriate adjustments. Colormaps specifically designed to accommodate a wide range of color vision deficiencies—such as the scientific colormaps developed by Fabio Crameri (Crameri et al., 2024)—could be a suitable option.

**Comment 2:** Please consider improving consistency in font family, font size, and colormap usage throughout the manuscript's figures. For instance, the labels in Figure 1c,d are difficult to read at their current size. In addition, the color shading in Figure 2 represents IVT magnitude but uses a different colormap than Figures 1 and 7. Unifying the colormaps would improve visual coherence across figures. Finally, in Figures 1 and 7, coastlines are less visible in panels showing total precipitation than in other panels, and increasing their contrast could improve readability.

**Line-by-line comments**

**Line 24:** *"…in the last about 60-year."* This phrasing is uncommon. Please consider changing it to *"…over the last ~60 years."*

**Line 29:** *"…relevant amounts of moisture…"*. Please specify that this refers to *atmospheric* moisture.

**Line 36:** Please define the acronym *CALJET*.

**Line 35:** *"Here, several field campaigns…"*. Please consider changing "Here" to "There".

**Lines 67-69:** Are there specific AR detection algorithms that are affected by such discontinuities and that the authors could explicitly reference?

**Line 75:** Please refer to the AR detection algorithm by Guan and Waliser as tARget rather than GW15, as this is its official name.

**Line 83:** Writing IVTx and IVTy is uncommon in geoscience. Please consider using IVTu and IVTv instead.

**Lines 89-90:** Please consider mentioning the number of area units in the first sentence: *"…the daily precipitation is aggregated over 94 area units of the Italian Department of Civil Protection, defined for the national operational warning system. These areas aggregate homogeneous subregional hydrological basins of the territory."*

**Line 126:** Please consider changing "infrastructures" to "infrastructure".

**Lines 129-130:** Please add the article before AR: *"The red bold line surrounds the object identified as an AR."*

**Line 133:** Please consider changing "expanded" to "increased".

**Lines 144-145:** *"However, other studies applied variable IVT thresholds based on the percentiles of IVT (e.g., 85th percentile, Lavers et al., 2012)…"* Please specify how the percentile is defined for that specific algorithm. Over which reference period is it calculated?

**Lines 153-154:** There is a typo: *"Contiguous areas that satisfy these criteria identify objects candidate to be ARs".*

**Line 155:** Please consider changing IVTy to IVTv.

**Lines 156-157:** *"IVT percentiles have been previously computed based on daily values at 12 UTC for the 30-year period ranging from 1991 to 2020."* Why are IVT magnitudes at 12 UTC used instead of daily IVT averages? Is this choice inherited from tARget?

**Line 159:** *"Based on the problems detected in its application to the two case studies described in the previous section…"* I could not find a description of such problems specifically related to tARget. Do the authors refer to the challenges mentioned in the introduction? If so, please clarify this point (see also my major comments).

**Lines 165-166:** *"…north-north-west to south-south-east…"* Is this a typo?

**Line 169:** Please add the article before AR and consider changing "values" to "magnitude": *"misses to recognise this object as an AR (Fig. 2a), although the IVT magnitude…"*

**Line 172:** Please consider changing IVTx to IVTu.

**Line 183:** *"(see text for more details)."* Please consider changing "text" to "Section 3".

**Lines 184-185:** *"for some of the 6-hour steps taken into consideration in the analysis of the ERA5 fields, the correct detection of the two ARs seems hampered…"* If I understand correctly, IVT percentiles are computed using only 12 UTC values, but are then applied to threshold the IVT magnitude at 00, 06, 12, and 18 UTC. If so, this may introduce a climatological bias related to the daily cycle of IVT.

**Lines 187-189:** *"After many trials and errors, undertaken by varying the pre-set values of these two parameters, the coherence is increased to 65° (instead of 45°), while the consistency check is discarded."* Is this new parameterization based on the two case studies of storms Alex and Vaia? How do these changes affect AR detection overall compared to the original tARget configuration?

**Lines 191-193:** *"Moreover, a recent release of the detection algorithm (Guan and Waliser, 2024, their Fig. 3 in particular) has included similar refinements, such as the possibility to detect zonal or even equatorward ARs."* If the authors are aware of the most recent release of tARget, why was this version not used as the basis for the analysis?

**Line 196:** There is a typo: *"Methodology for the identification of intense ARs affecting northern-central Italy"*

**Line 200:** *"…and group together those related to the same event."* I assume the authors refer here to the tracking of AR trajectories. If so, a description of the AR tracking strategy appears to be missing from the manuscript.

**Line 202:** There is a typo: *"of a southerly low-level jet"*

**Line 206:** *"an almost straight line."* Please consider rephrasing, as AR axes are rarely close to straight.

**Line 206:** *"Thus, it may happen that "false" ARs are revealed."* Please consider changing "revealed" to "detected".

**Line 208:** Please consider changing "landing" to "making landfall".

**Lines 212-214:** *"Also, a temporal requirement is imposed in the AR detection, that is the AR must cover the target area and the remote source regions in Fig. 3a for at least three 6-hourly time steps.*

*It means that only AR objects persisting longer than 12 hours are considered as AR events, as set also in previous studies"* This implies an AR tracking strategy that is not described in sufficient detail in the manuscript.

**Line 213:** *"…for at least three 6-hourly time steps. It means that only AR objects persisting longer than 12 hours…"* Please consider changing "longer than 12 hours" to "for 18 hours or more".

**Line 223:** *"a partial failure of the algorithm in computing the geometrical requirements"* Please clarify why the algorithm partially fails to compute the geometrical requirements.

**Figure 3:** A colorbar for the shading representing orographic elevation is missing.

**Line 234:** A comma is missing: *"During 1987, a maximum…"*

**Figure 4:** The results shown in this figure may be highly sensitive to the AR detection algorithm used, further highlighting the need for an uncertainty assessment.

**Line 258:** *"al. (2020) that studied extreme precipitation over the same target area."* Please consider changing "that" to "who".

**Table 1:** The last column appears to result from applying the AR strength scale to ARs rather than to AR conditions. Please revise.

**Line 293:** There is a typo: *"…connected with ARs."*

**Line 298:** There is a typo: *"…and ARs are…"*

**Line 340:** There is a typo: *"…SD and DM. All authors…"*

**References**

Crameri, F., Shephard, G. E., & Heron, P. J. (2024). Choosing Suitable Color Palettes for Accessible and Accurate Science Figures. *Current Protocols, 4*(8), e1126.

Guan, B., & Waliser, D. E. (2015). Detection of atmospheric rivers: Evaluation and application of an algorithm for global studies. *Journal of Geophysical Research: Atmospheres, 120*(24), 12514-12535.

Guan, B., Waliser, D. E., & Ralph, F. M. (2023). Global application of the atmospheric river scale. *Journal of Geophysical Research: Atmospheres, 128*(3), e2022JD037180.

Guan, B., & Waliser, D. E. (2024). A regionally refined quarter-degree global atmospheric rivers database based on ERA5. *Scientific Data, 11*(1), 440.

Ralph, F. M., Rutz, J. J., Cordeira, J. M., Dettinger, M., Anderson, M., Reynolds, D., Schick, L. J. & Smallcomb, C. (2019). A scale to characterize the strength and impacts of atmospheric rivers. *Bulletin of the American Meteorological Society, 100*(2), 269-289.

Vallejo-Bernal, S. M., Braun, T., Marwan, N., & Kurths, J. (2025). Pikart: a comprehensive global catalog of atmospheric rivers. *Journal of Geophysical Research: Atmospheres, 130*(15), e2024JD041869.

---

## Author Comment (AC1)

We would like to thank the Reviewer for the careful reading of the manuscript and for the valuable comments and suggestions. We found them very constructive, and we will certainly do our best effort to address all of them, as detailed below.

*1) This manuscript focuses on a specific type of AR intrusion from the North Atlantic into the western Mediterranean, reaching northern–central Italy. While this focus is generally well explained throughout the manuscript, it may be helpful to occasionally remind the reader that these events are not representative of ARs across the entire Mediterranean basin (e.g., line 327). I would suggest considering the addition of "the western Mediterranean" to the title. Moreover, it might be beneficial to ensure that this scope is stated consistently throughout the manuscript, including in the abstract, for example as it is clearly expressed in the first sentence of the Conclusions (lines 306–307)*

We agree with this comment. We are aware that our analysis only includes ARs over the western Mediterranean and in this sense, it is even conservative, because it focusses only on those ARs that reach the target area of northern-central Italy. We will stress this better, and we accept the suggestion for modifying the title. However, we would like to point out that the analysis is not limited to ARs from the North Atlantic, because a number of events are characterized by transport from Tropical Atlantic, across Africa.

*2) Other studies reported similar behavior of ARs reaching eastern Mediterranean and middle east, I think those could be included in the introduction (lines 49-61) as these investigate the eastern side of the same Mediterranean basin. For example:*
*- Francis, D., et al. (2024) Atmospheric river rapids and their role in the extreme rainfall event of April 2023 in the Middle East. Geoph.Res. Lett., 51, e2024GL109446.*
*- Ezber, Y., et al. (2024) Impact of atmospheric rivers on the winter snowpack in the headwaters of Euphrates-Tigris basin. Clim Dyn 62, 7095–7110 (2024).*

We are aware of the interesting studies of Francis, and one is already cited in the Introduction. We will add also these two references as they pertain to the literature of Mediterranean ARs.

*3) Could the authors clarify why extreme precipitation events are selected using the 99th percentile computed only over the climatic period 1991–2020, rather than over the full dataset period (1961–2024)? If extreme precipitation has increased over recent decades, using the 99th percentile based on the last 30 years could potentially lead to missing extreme events from the earlier part of the record. An alternative approach could be to de-trend the time series and then identify extreme precipitation events based on the de-trended data. Even if the differences in the selected extreme events are small, please ensure that this choice does not affect the robustness of the results.*

For the precipitation, we exploited the analysis previously produced by Grazzini et al (2024), who aggregated the rainfall on areas used operationally for the national warning system of the civil protection. This approach has several benefits: it allows to aggregate rainfall on subregional hydrological basins, which are climatologically homogenous; with this upscaling approach, localized events smaller than roughly 300 km$^2$ are disregarded; it allows to keep a strong link with operational applications. Thus, following the analysis carried out by Grazzini et al (2024) and sharing the same

philosophy, performing the 99[th] percentile computation on the recent 30-year period 1991-2020 is aimed at reaching results almost applicable to operations, since we are able to recognise EPE with respect to recent climatic conditions.

In any case, we will check to what extent this choice influences extreme events in the past

Grazzini, F., et al: Improving forecasts of precipitation extremes over northern and central Italy using machine learning. Q. J. R. Meteorol. Soc., 150, 3167–3181, https://doi.org/10.1002/qj.4755, 2024.

*4) In lines 156-157 similar concern arises on why the IVT 85th percentile is calculated using the period of 1991 to 2020 and not the full length of the data. As before please make sure this does not affect your results*

We believe that 30y period is long enough to capture the climate of IVT in the Mediterranean. In the literature, much shorter periods are usually considered. Moreover, the monthly values are used by the detection algorithm to compute a 5-month centred average, thus IVT values become even smoother. The final values used as threshold in the algorithm are relatively low with respect to the typical IVT values that are attained in the area when weather systems force WV transport. Therefore, we are very confident that the selected period does not affect our results.

In any case, we checked for the month of September, which presents the highest values of IVT, and the difference between the values corresponding to the 85[th] percentile computed in 30 or in 60 years have a quite random pattern and a limited magnitude, being always below than 10% (rarely exceeding 20 kg/m/s). The same check has been done also for a winter month (January).

*5) In lines 187–189, does this mean that the difference between the two directions (mean IVT and object orientation) is allowed to be as large as 65°? The wording was somewhat confusing when referring to "coherence." Given that these adjustments to the GW15 algorithm are extremely valuable for the scientific community interested in applying AR detection methods to regions with complex orography, it would be helpful to clarify them as much as possible. The rest of the modifications applied are reasonable, have been tested and seem to work well.*

Coherence checks the grid cell IVT direction with respect to the object mean IVT. Concerning coherence, the change was required for AR presenting a particularly marked U shape, moving SE-ward over the Atlantic and NE-ward over the Mediterranean. With a marked U shape, the mean IVT is longitudinally oriented, but a relevant part of the AR presents a meridional orientation exceeding 45°, thus being discarded. In few (but relevant as for the 2020 event) cases, relaxing the coherence check allows to detect much better the ARs entering the Mediterranean.

The other important modification concerned the direction of transport, allowing southward IVT. Both these refinements have been taken into consideration in the last release of the code (as mentioned in the text).

We will further check the consistency requirement, which compare the object mean IVT direction and the overall orientation, to filter objects where the IVT does not transport in the direction of object elongation. In fact, it does not seem to be necessary to change this requirement (it was just a preliminary test before identifying coherence as the most sensible parameter).

*6) In lines 219-221, it is not clear where the maximum-IVT is measured and how this location is selected. Is it a fixed point in the Ligurian sea for all ARs?*

That's right. We will specify it. The max IVT is detected over the Ligurian or the Tyrrhenian Sea (see Fig. 3), before reaching the coast of the considered target area, thus over the sea grid-points between 40-44.5° N and east of 7°E.

*7) Together with Figure 3, I would have appreciated seeing the AR frequency climatology for ARs entering the western Mediterranean and reaching northern–central Italy. This would allow for a quick assessment of the AR detection methodology proposed in the manuscript and facilitate comparison with other algorithms. For instance, based on my experience, detecting ARs using global detection algorithms within a limited domain (20–60° N; 30° W–30° E) can sometimes lead to missed ARs near the domain boundaries, as those might be partially outside of the domain. This may not be a major issue in the present case, since only ARs that reach northern Italy and the arc-shaped region are considered; nevertheless, including the AR frequency would still be useful to visualize the spatial distribution of the detected events.*

We did not include the requested picture in the original manuscript, since we thought it would have not added much information. However, we are happy to add in the revised version.

[Figure]

The figure shows the number of 6-h timesteps during which a grid point is within the shape of an AR. Given the relatively small target area of norther-central Italy, there is a clear convergence of the detected ARs towards it. It clearly highlights two main AR pathways, one from the Atlantic, the other from North Africa.

*8) Figure 4 shows the number of events per year, and from a visual inspection there appears to be a possible positive trend. Could the authors clarify whether this trend reflects a warming climate effect or whether it might instead be an artifact of defining the IVT threshold using the 85th percentile from*

*the most recent period? It would be helpful to verify whether this behavior is sensitive to the AR selection methodology (see my comment #4), and, if not, to assess and report whether the trend is statistically significant.*

The computation of a possible trend in the AR number provides a value of +0.03 events per year, so a very weak increase ($R^2$= 0.06) and not statistically significant (p=0.07).
As in the reply to the comment #4 above, the computation of IVT 85th percentile over a 30-year period should not impact the analysis.

*9) In line 250, the AR scale by Ralph et al. (2018) is used. It would be helpful to acknowledge that this scale was originally developed for the west coast of the United States and is therefore not necessarily tailored to ARs making landfall within the Mediterranean basin. Nevertheless, the scale remains useful and relevant for presenting the results shown here. One possible option could be to apply the scale to the arc-shaped area outside the Mediterranean in order to assess the intensity of the ARs before they enter the basin.*

We agree with this comment, and we will add it in the text.
We do not believe that applying the scale on the arc-shaped area outside the Mediterranean is useful, since very different conditions characterize this area: open ocean in the west, desert in the south. In the literature, the impact of an AR is always connected to its intensity close to landfall, and that's what we are interested in.

*10) In line 308, it might be worthwhile to briefly restate the main modifications made to the GW15 algorithm for application in the Mediterranean basin. I believe this represents an important outcome of the study and would be well suited for inclusion in the Conclusions.*

We will recall this in the conclusions. However, we are also aware that these modifications are not relevant or have been already implemented in the last version of the algorithm which has been recently released.

*11) In line 326, the manuscript refers to the interesting results from Mastrangelo et al. (2025). It might be helpful to briefly summarize the nature of these results for the reader. In addition, since the current reference appears to point to a conference abstract, the authors could consider citing a preprint of this work, if available.*

Unfortunately, a preprint is not available yet, since it is still work in progress. Anyway, we will elaborate the sentence better, providing more details concerning both our study and recent literature dealing with the sub-seasonal range.